# Deciphering Self-Improvement: Large Language Models Can Take False First Steps

## Abstract

One of the most striking capabilities of Large Language Models (LLMs) is their apparent ability to refine outputs through a process of self-improvement. Yet how an autoregressive model acquires such capability remains unclear. We propose a mechanistic model of LLM self-improvement grounded in a Bayesian perspective on token generation. In this view, LLMs maintain latent plans for future token generations, which gradually stabilize as more self-generated tokens are autoregressively incorporated back into the context. Across two single-dimensional random number generation experiments, we find evidence consistent with the dynamic patterns of planning and generation predicted by this Bayesian model. Building on these insights, we introduce self-play Markov Chain Monte Carlo (spMCMC), an extension of MCMC-with-LLMs designed to elicit reliable reward signals for self-improvement in open-ended text generation. Human evaluations show that spMCMC identifies higher-quality outputs that are often overlooked by both greedy decoding and other self-evaluation methods.

## 1 Introduction

Recent work has increasingly explored the self-improvement capabilities of large language models (LLMs), wherein LLMs refine their outputs through iterative generation and self-evaluation without additional supervision (Zelikman et al., 2022; Xie et al., 2023; Tian et al., 2024; Park, 2025). While significant gains have been demonstrated in diverse domains (Zelikman et al., 2022; Ren et al., 2023; Piché et al., 2024), the underlying mechanisms driving these improvements remain poorly understood. Existing studies have focused either on documenting empirical performance improvements or on developing principled frameworks to structure the process (Huang et al., 2024; Song et al., 2024), but offer limited insight into how LLMs internally derive the reward signal underpinning self-improvement. Although much of the literature is motivated by the intuition that "evaluation is easier than generation," none has directly addressed the question of why does a gap exist between generation and evaluation in LLMs that are trained for generation?

There is a useful parallel in human cognition. When given an instruction, humans often find it easier to evaluate their own linguistic outputs (e.g., reviewing a recorded speech or rereading a written text) than to produce them in the first place (Lind & Hartsuiker, 2020). This asymmetry arises because production requires coordination of complex biological systems beyond merely perception (e.g., Broca's area, motor cortex, musculoskeletal system) (Broca, 1861). By contrast, LLMs, despite being trained on massive quantities of human language, differ fundamentally from humans. LLMs generate text auto-regressively without an explicit perceptual system operating separately from the production mechanism. Consequently, there is no inherent reason to expect LLMs to evaluate better than they generate, since both processes rely on the same neural information processing system.

Re-examining how LLMs generate, a recent line of work has highlighted their planning capabilities (Pal et al., 2023; Men et al., 2024; Pochinkov et al., 2024; Wu et al., 2024; Jenner et al., 2024; Lindsey et al., 2025; Cencerrado et al., 2025). LLMs can be seen as implicitly forming latent plans for their responses before generation unfolds, with intermediate hidden representations encoding sequence-level regularities. In this study, we argue that although LLMs are capable of planning ahead, their token-level generations often deviate from these sequence-level plans, especially at the onset of generation when outputs are strongly biased by token priors without context (McCoy et al., 2024; Bachmann & Nagarajan, 2024). These biased early tokens subsequently drive adjustments

to the model's planning until generation converges toward a stable distribution aligned with the plan. This recursive process creates drift between latent sequence-level plans and realized outputs, as small deviations at the start can cascade and reshape later continuations. We propose that this reflects an inherent tension between implicit planning and token-level realization in auto-regressive LLMs. During this transition, generations may remain semantically unstable prior to convergence, potentially lowering overall output quality. As a result, this gap provides the opportunity for self-improvement.

Building on the gap between LLMs' planning and generation, we propose a Bayesian model to explain the emergence of reward signals for LLM self-improvement (see Figure 1). In the following sections, we formalize this model and empirically validate it through a series of experiments. Building on these insights, we introduce a novel method, called *self-play Markov Chain Monte Carlo* (spMCMC), inspired by techniques for eliciting human mental representations to recalibrate LLM outputs based on their implicit plans. Our results show that LLM-generated completions reweighted by spMCMC correlate positively with human ratings.

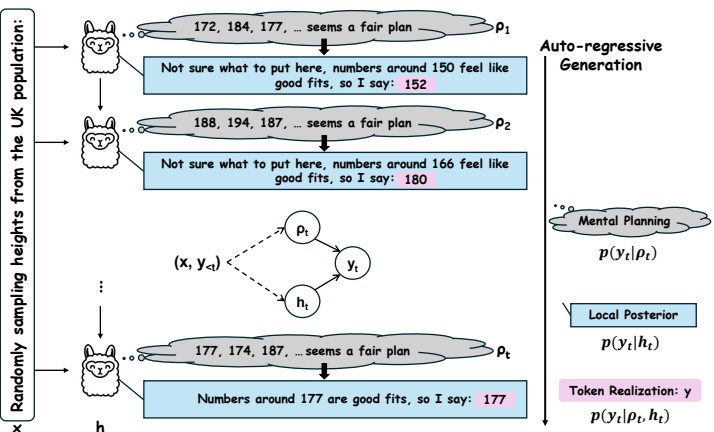

Figure 1: The Bayesian model of token generation predicts a dynamic interaction between planning and local inference for autoregressive generation. At the start of generation given a task prompt $x$, the next-token preference is highly uncertain, making local inference particularly susceptible to token-level priors that may diverge from the model's latent plans. As self-generated tokens accumulate and uncertainty diminishes, local inference becomes less susceptible to priors and progressively aligns with the latent plans, which in turn becomes increasingly dominant in shaping subsequent generations.

## 2 RELATED WORK

Self-improvement is a reliable method to bootstrap the performance of LLMs. In a typical self-improvement pipeline, an LLM first generates multiple responses to a given query, then refines its own outputs through self-evaluation such as voting (Huang et al., 2022), scoring (Liang et al., 2024), sampling (Zelikman et al., 2022; Song et al., 2024; Wang et al., 2022) or other forms of feedback (Yuan et al., 2024). Significant performance improvements have been observed in inference-time refinement or supervised fine-tuning (SFT) based on this paradigm (Ren et al., 2023; Piché et al., 2024). Building on this empirical evidence, Huang et al. (2024) proposed a mechanism termed 'sharpen', which explains LLMs' capacity for self-improvement during post-training by concentrating probability mass on high-quality sequences. Song et al. (2024) further developed a principled framework to formalize the self-improvement governed by a generation-verification gap.

Understanding the origin of this gap requires theorizing about LLMs' process of generating tokens. From a Bayesian perspective, human language can be modeled as a probabilistic generative process, which starts from priors over sequence-level representations, e.g., syntax, grammars, semantics (Chater & Manning, 2006). Generation of responses $y$ to the context $x$ then can be described

as inferring structured regularities based these priors and making prediction. In contrast, LLMs encode human language by learning to maximize next-token likelihood (Radford et al., 2018). On one hand, this makes LLMs go beyond pure improvisation by capturing implicit sequence-level regularities within single-token inference. Empirical studies have demonstrated such planning capacities of LLMs in intermediate layers (Pochinkov et al., 2024; Wu et al., 2024; Jenner et al., 2024; Lindsey et al., 2025), particularly for short-term future generations (Pal et al., 2023; Men et al., 2024), where models appear to plan ahead before actual generation begins. Consequently, future outputs can be decoded from the embeddings of earlier tokens (Men et al., 2024; Pochinkov et al., 2024). On the other hand, LLMs acquire extensive prior knowledge from human language, which serves as the foundation for their behaviors (Zhang et al., 2024; Zhu & Griffiths, 2024; Arai et al., 2025; Ban et al., 2025), belief updating (Imran et al., 2025) and even biases (Chen et al., 2023; McCoy et al., 2024).

Moreover, recent work has shown that errors accumulate during extended sequential token generation, often leading to degraded outputs such as hallucinations (Kalai et al., 2025). Another relevant line of work highlights the inherent differences between the distributions of human- and LLM-generated data, both in specific tasks (Harrison, 2024) and in broader text generation (An et al., 2024; Zhu et al., 2024b). This suggests that even when trained extensively on human data, the generative process of LLMs introduces systematic deviations from the training distribution, yielding stylized outputs that enable a high rate of self-recognition (Panickssery et al., 2024).

## 3 A THEORETICAL FRAMEWORK OF LLM SELF-IMPROVEMENT

**A Bayesian model of token generation in auto-regressive models.** LLMs typically use the decoder architectures of the Transformer (Vaswani et al., 2017), which are trained to minimize the negative log likelihood of next tokens using the pretraining dataset $\mathcal{D} = \{d_{1:Tn}^{(n)}\}_{n=1}^N$: $\mathcal{L}_{NLL}(\theta) = -\sum_{n=1}^N \sum_{t=1}^{Tn} \log p_\theta(d_t|d_{<t})$. As such, LLMs can learn the distribution of the next token conditioned on the previous ones. Analogously, during inference-time given a query $x$, LLMs will respond $y$ following the distribution: $p_\theta(y|x) = \prod_{t=1}^T p_\theta(y_t|x, y_{<t})$.

Unlike Bayesian language models, LLMs' auto-regressive production proceeds without explicitly inferring any latent representations (Radford et al., 2018). However, it can be learned as chained mappings on the latent states $\mathbf{h}$ conditioning the next token at each generation step. Given a prompt $x$, the distribution of responses $y$ then can be factorized as the following two-stage transformation:

$$p(y|x) = \prod_{t=1}^T p(y_t|\phi_t), \ \phi_t = f(x, y_{<t}) \tag{1}$$

where $\phi_t$ represents intermediate neural activations given the current context $(x, y_{<t})$, containing information about LLMs' comprehension of the context (i.e., $\mathbf{h}_t$) and the latent plan for future generation not yet occurred (i.e., $\rho_t$). These representations then drive token generation through $p(y_t|\phi_t) = p(y_t|\rho_t, \mathbf{h}_t)$. Therefore, we effectively assume the generative structure of token generation as shown in Figure 1:

At each $t$, LLMs generate the token $y_t$ based on planning $\rho_t$ and contextual representation $\mathbf{h}_t$ given the query and previous generations $(x, y_{<t})$. Applying the Bayes' rule, we obtain the following relationship:

$$p(y_t|x, y_{<t}) = p(y_t|\rho_t, \mathbf{h}_t) \propto \underbrace{p(\rho_t|y_t)}_{\text{planning}} \cdot \underbrace{\underbrace{p(\mathbf{h}_t|y_t)}_{\text{local likelihood}} \cdot \underbrace{p(y_t)}_{\text{prior}}}_{\text{local posterior inference}} \tag{2}$$

The planning term $p(\rho_t|y_t)$ measures how well a candidate token $y_t$ aligns with the intended plan (i.e., $\rho_t$). In other words, this term determines how instruction-following the generation is as planning is calibrated on sequence level. In contrast, the improvisational likelihood, $p(\mathbf{h}_t|y_t)$, focuses on coherence locally, measuring how compatible token $y_t$ is with the observed history, without considering future planning. Finally, the unconditional prior, $p(y_t)$, reflects model's preference for the token $y_t$ given no context, accordingly serving as a source of bias that may jeopardize coherence and instruction-following (c.f., McCoy et al., 2024). Therefore, the token generation process could

be viewed as an interaction between local inference and planning: the token agreed by both local posterior inference and sequence-level planning will be generated.

**Predicted dynamics of token generation with planning.** How does the Bayesian model of token generation evolve during token-by-token decoding? Because LLMs repeatedly recondition on their expanding sequence of generated tokens, the dynamics can be driven by an asymmetry: the model processes the query, $x$, differently from its own responses, $y_{<t}$. A growing body of work on LLM-as-a-judge has demonstrated a strong self-preference bias: models prefer their own generations and tend to overestimate their quality (Wataoka et al., 2024; Panickssery et al., 2024; Chen et al., 2025). This arises because an LLM's outputs are, by construction, highly consistent with its learned conditional probabilities, making them appear more "familiar" to the model than semantically-similar text produced by other models.

We therefore hypothesize that high perplexity (i.e., less familiarity) of user query or instruction at the start of generation — where the model must condition primarily on the out-of-distribution (OOD) input — elevates the uncertainty in token-level prediction. Concretely, this leads to a flatter distribution for the context compatibility $p(\mathbf{h}_t|y_t)$. As a result, local inference of the token posterior distribution will be more strongly influenced by the unconditional prior $p(y_t)$. Thus, any mismatch between the local token posterior and the planning term leads to a weaker alignment, inducing biased token generation.

In the next step, the sequence evolves from $(x, y_{<t})$ to include the biased, self-generated token $y_t$, resulting in the new context $(x, y_{\leq t})$. This new context then drives the update on the plan $\rho_t$ by updating the neural representations from $\phi_{t-1}$ to $\phi_t$. As generation proceeds, the proportion of OOD information in the query as part of the context diminishes, causing decreasing uncertainty in local likelihood $p(\mathbf{h}_t|y_t)$. That is, the LLM becomes progressively more confident about which tokens are compatible with the current context. Moreover, local likelihood also becomes increasingly aligned with planning $p(\rho_t|y_t, \mathbf{h}_t)$, as both are aware of joint context now. Together, the gradual inclusion of biased tokens generated by the LLM itself should suppress the influence of the unconditional prior $p(y_t)$ (i.e., the source of the bias), leading to more coherent and instruction-following outputs. Ultimately, the close alignment between planning and local inference enables converged and coherent token generation that continues with respect to the same plan.

**Predicted signals for the self-improvement.** According to the Bayesian model, the signals that enable self-improvement fundamentally arise from the process of converging to a plan. Self-evaluation relies on incorporating self-generated outputs directly into the prompt. As LLMs need some time to align their generation with planning, tokens generated initially are more likely to be incoherent with respect to the converged plan. Including self-generated content in the prompt (i.e., locally-coherent tokens) provides an option to accelerate convergence, effectively "resetting" the process midway and reducing incoherence or irrational output that may appear at the onset of generation. Thus, the outputs in self-evaluation capture a reward signal relative to the original outputs.

## 4 EXPERIMENTS

### 4.1 EXPERIMENT 1: RANDOMLY GENERATING HUMAN HEIGHTS

To quantify the dynamics of planning in sequential token generation, we adopt a random number generation task from Castillo et al. (2024). The task was chosen for its simplicity, as it produces outputs in a one-dimensional numeric space where predictive relationships can be easily measured. Using open-source LLMs (Llama-3.1-8B-Instruct and Qwen-2.5-7B-Instruct), we elicited random numeric estimates of human height to probe how models form and commit to future answers in advance. What makes this task appealing for evaluating plans is that the models are prompted to produce independent random estimations consecutively, thereby minimizing confounds from auto-correlated continuations within sequences that LLMs may produce (see prompt in Appendix A.1).

The temperature of generation is set at 0 (greedy decoding) to maximize the planning behavior. In each trial, LLMs generated 60 estimates in sequence, with each pair of consecutive outputs separated by a comma and a space (e.g., "182, 163, 159, ..."). Each model completed 69 trials, each initialized with a unique starting value ranging from 151 cm to 220 cm. Thus, every model produced 69 distinct sequences of height estimates (see Figure 6 in Appendix), with each sequence beginning

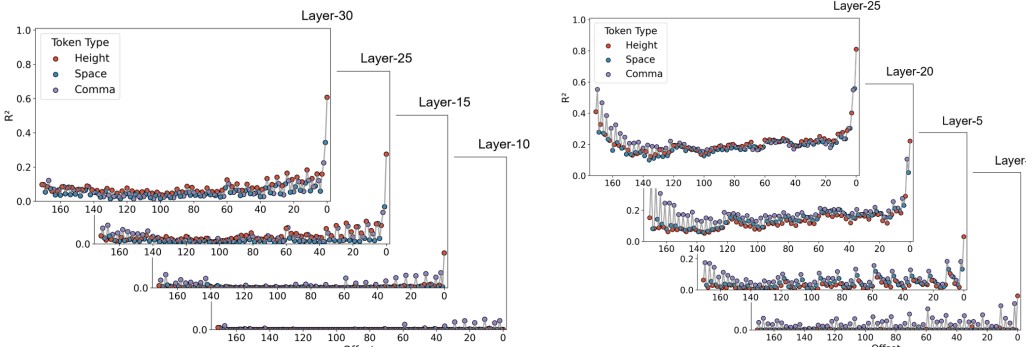

Figure 2: Percentage of variance in future tokens ($R^2$ of regressions; vertical axis) captured by the LLM embeddings across different planning horizons (offset $\Delta t$; horizontal axis): **(left)** Llama-3.1-8B-Instruct, **(right)** Qwen-2.5-7B-Instruct.

with a number between 151 and 220 and containing a total of 60 values. Beside the completions, the embeddings ($d = 3584$ for Qwen and $d = 4096$ for Llama) for all new tokens from both models were also collected to reflect the information on which each new estimate was built.

**Data Analysis.** We evaluate how well the LLM embedding extracted at time $t$, $\phi_t$, predicts future tokens $y_{t+\Delta t}$ with a look-ahead offset of $\Delta t$. This analysis has two implications. First, if $\phi_t$ is predictive of $y_{t+\Delta t}$, it suggests that the contextual representation and the plan influence future token generation. Under the correct plan of random number generation (i.e., when the LLM strictly follows the user instruction), however, $\phi_t$ should not be related to $y_{t+\Delta t}$, since each token is expected to be generated independently at random. Second, we are also interested in how the offset $\Delta t$ modulates the predictive relationship between $\phi_t$ and $y_{t+\Delta t}$. If smaller offsets correspond to stronger relationships, this would suggest that LLM embeddings may converge toward specific plans.

We applied LASSO regression with a penalty coefficient $\alpha$ as 0.3 to estimate the prediction performance of embeddings $\phi_t = f(x, y_{<t}) \in \mathbb{R}^{n \times d}$ in forecasting a future response value $y_{t+\Delta t} \in \mathbb{R}^n$ ($\Delta t \geq 0$). The offset parameter $\Delta t$ ranged from 0 to 172, reflecting the lag between embeddings and subsequent numeric outputs (corresponding to a prediction horizon of 0–58 numeric samples, as commas and spaces were included in the tokenization). For each offset, all available training data $\{\phi_t^{(n,:)}, y_{t+\Delta t}^{(n,:)}\}_{n=1}^{69}$ from 69 sequences were extracted and aggregated, and a separate regression model was fitted to evaluate the degree to which embeddings at that offset encode information about upcoming responses.

$$y_{t+\Delta t} = \phi_t \omega_{\Delta t} + b_{\Delta t}, \ 0 \leq \Delta t \leq 172 \tag{3}$$

Considering the embeddings as the comprehension of current context and prediction, the regression coefficients $\omega_{\Delta t} \in \mathbb{R}^d$ on future values then could represent the planning behavior at given positions. The corresponding fitting goodness, $R^2$, indicates the strength of the planning across specific planning horizon.

Then, to further examine how planning strength evolves over the course of generation, we fixed the planning horizon at $\Delta t = 8$, which provides a fair window for quantifying predictive planning. At each comma position $q$, we constructed datasets $\{\phi_q^{(n)}, y_{q+8}^{(n)}\}_{n=1}^{69}$ and fitted a separate LASSO regression model. This allowed us to track how well embeddings eight steps beforehand anticipate responses at position $q$, thereby revealing the dynamics of planning signals across the sequence.

$$y_{q+8} = \phi_q \omega_q + b_q, \ q = 3i + 1, \ 0 \leq i \leq 57 \tag{4}$$

**Results.** In the random generation task, positive percentages of variance in future tokens captured by the embeddings suggest the existence of planning behavior across almost all layers of both LLMs (see Figure 2). In particular, both LLMs begin planning for future token generations from the early layers at the comma positions following each sample. At the sample positions themselves, the models first reflect on the current token in the early to middle layers before planning ahead in the later layers. Notably, the strength of planning decreases as the offset increases, indicating that the plan shifts during generation.

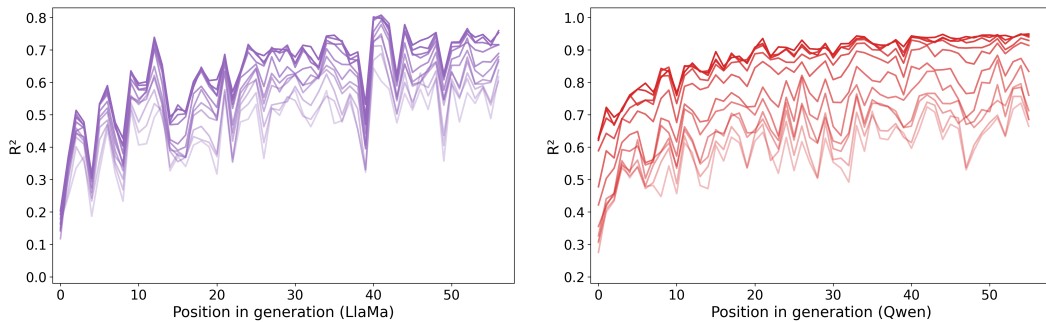

Figure 3: Relationships between plan adherence and planning position: **(left)** Llama-3.1-8B-Instruct, layer 15-25 from lightest to darkest, **(right)** Qwen-2.5-7B-Instruct, layer 10-20 from lightest to darkest. Higher $R^2$ indicates stronger adherence of later token generations to the latent plans formed eight steps earlier.

The dynamics of planning strength during generation is shown in Figure 3, revealing a consistently upward trend of prediction performance $R^2$ as generation proceeds across all mid-layers. Compared to early planning, generation is more closely aligned with later planning, reflecting greater convergence over time.

## 4.2 EXPERIMENT 2: RANDOMLY GENERATING NUMBERS FROM GAUSSIAN DISTRIBUTIONS

Based on the Bayesian model of token generation, a user query $x$ — being likely OOD for LLMs — would typically induce a bias toward the prior token distribution $p(y)$. However, as the LLM generates more locally coherent tokens (via local posterior inference), the updated context $(x, y_{<t})$ reduces uncertainty in subsequent token generations. This process can be characterized as biasing-then-debiasing.

To test whether conditioning on self-generated tokens, $y_{<t}$, facilitates the debiasing of future tokens, we conducted a second experiment in which, as in Experiment 1, LLMs were prompted to generate random numbers–this time sampled from Gaussian distributions, $\mathcal{N}(\mu, 10)$. We first generated 64 random samples from the Gaussian distribution and then prompted the LLM to continue the sequence with an additional 64 samples (i.e., Gen I samples in Figure 4). Note that the LLMs were not provided with the information about the Gaussian distribution (see Appendix A.2 for detailed prompts). Next, we conditioned the LLM only on the 64 self-generated samples (i.e., Gen I samples in Figure 4) and again prompted it to continue generating random numbers (i.e., Gen II samples in Figure 4). This design yielded two comparison groups: one beginning with 64 random samples from the Gaussian, and the other beginning with 64 samples generated by the LLM.

**Data Analysis.** For this task, the rational plan $\rho$ is to generate numbers randomly from $\mathcal{N}(\mu, 10)$, although the LLMs must infer the Gaussian distribution from the observed context samples. Seven different $\mu$ values, ranging from $-50$ to $50$, were tested in the experiment. Each of the seven conditions was repeated 100 times, with a different set of 64 randomly generated samples prepended in each run. By examining the distributions of LLM-generated outputs at each step within each condition, we compared them against the ground truth (the true conditional distribution, shown as starting squares in Figure 4) and recovered the dynamics of output quality over the course of generation.

**Results.** As shown in Figure 4, consistent starting patterns emerged across the seven conditions. For both LLMs, the initial generations were biased toward negative values (around -30) before being de-biased — sometimes even over-corrected — eventually converging to stable distributions that closely match the mean of the underlying Gaussian. This biasing–then–debiasing pattern aligns with the prediction of the Bayesian model of token generation when the initial random samples $y_{\leq 64}$ were not produced by the LLM (see statistics in Appendix C).

However, when conditioned on self-generated samples, the subsequent generations (i.e., Gen II samples) exhibited less bias compared to Gen I samples, which were conditioned on externally

provided random samples from the Gaussian distribution. This effect can also be explained by the Bayesian model: because the conditioning tokens $y_{\leq 64}$ are generated by the LLM itself, they are more in-distribution, thereby reducing the initial bias.

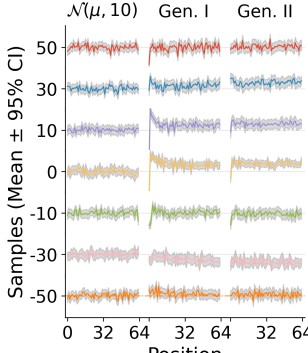 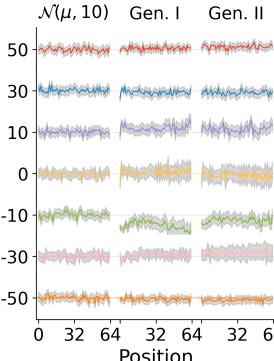

Figure 4: Generation start-off dynamic patterns in predictive sampling task: **(left)** Llama-3.1-8B-Instruct, **(right)** Qwen-2.5-7B-Instruct. Both panels display the mean with 95% confidence intervals of samples at each generation position, with the seven conditions presented vertically. Within each condition, three sets of samples are shown from left to right: (i) samples from a Gaussian distribution $\mathcal{N}(\mu, 10)$, (ii) samples generated by the LLM conditioned on the Gaussian (Gen. I), and (iii) samples generated by the LLM conditioned on Gen. I (Gen. II). We find statistically significant negative biases in the initial samples of Gen. I (see Tables 2 and 3 for details).

### 4.3 EXPERIMENT 3: MARGINALIZING OVER DIVERSE LATENT PLANS FOR BETTER EVALUATIONS

By limiting the LLMs output into a one-dimensional number line in Experiments 1 and 2, we observed a gradual stabilization due to the interplay between planning and generation. This converging process effectively provides a reward signal that incentivizes self-improvement in LLMs.

Here, we build on these findings to recover text generations that more closely align with a predefined plan: generating funny jokes. Users may instruct an LLM to produce funny jokes (i.e., $x$), but there is no guarantee that the model's latent plan $\rho$ and its generated text $y$ will align with the query perfectly, since funniness is inherently high-dimensional and multifaceted. For instance, the realized plan might emphasize insulting humor, jokes that rely on specific background knowledge, or more family-friendly forms of humor. As a result, the Bayesian model predicts that LLMs would generate texts that are contingent on a specific latent plan: $\{(y_i, \rho_i)|x\}_{i=1}^N$.

When prompting the same LLM to evaluate its own generations, the generated evaluations are actually adapted to the stabilized plans that are implicitly defined in its own generations. As such, we hypothesize that the model evaluates outputs on the basis of $Q(y_i, \rho_i \mid x)$ (that is, higher $Q$-values correspond to higher ratings). In other words, the LLM judges a generation $y_i$ jointly with its latent plan $\rho_i$. Common LLM-as-a-judge methods (e.g., rating and importance sampling) are therefore better understood as approximating $Q(y_i, \rho_i \mid x)$. By contrast, users are typically interested in the marginalized evaluation $Q(y_i \mid x)$. To marginalize over latent plans while retaining the LLM-as-a-judge framework, we introduce a new MCMC-with-LLM method that incorporates self-play.

**Self-play Markov Chain Monte Carlo (spMCMC) with LLM.** spMCMC is an extension of the behavioral elicitation methods of MCMC with People (Sanborn et al., 2010; Yan et al., 2024) and MCMC with LLM (Zhu et al., 2024a). The method proceeds in two stages. In the random generation stage, LLMs are encouraged to output as many as random answers $y$ converged on corresponding plans $\rho$ given a user query $x$. We set the temperature of LLMs at 1.0; that is, sampling from $(y, \rho) \sim p_\theta(\cdot|x)$. Then, we compute the pairwise cosine similarity between any two text generations (i.e., $y_i$ and $y_j$) by embedding them using a Sentence BERT model $f_{bert}$ (Reimers & Gurevych, 2019):

$$S_{ij} = \frac{\mathbf{e}_i \cdot \mathbf{e}_j}{\|\mathbf{e}_i\|_2 \|\mathbf{e}_j\|_2}, \; \mathbf{e}_i = f_{bert}(y_i) \; \text{and} \; \mathbf{e}_j = f_{bert}(y_j) \tag{5}$$

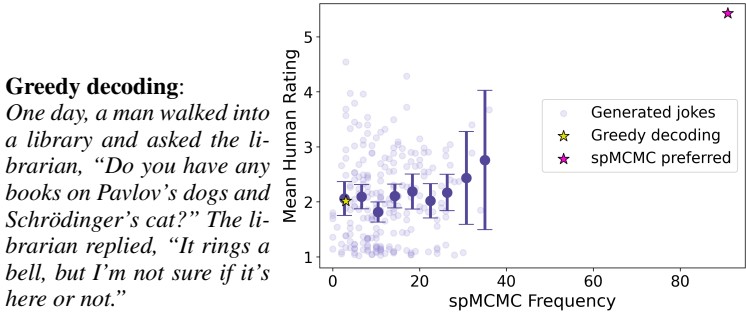

**Greedy decoding**:
*One day, a man walked into a library and asked the librarian, "Do you have any books on Pavlov's dogs and Schrödinger's cat?" The librarian replied, "It rings a bell, but I'm not sure if it's here or not."*

★ **The funniest joke as ranked by spMCMC**:
*I told my wife she was drawing her eyebrows too high. She looked surprised.*

Figure 5: Relationship between spMCMC-derived joke frequency and human-assessed quality of the jokes. Each dot denotes a joke. Error bars represent the window-binned means with $\pm$ 95% confidence interval of the underlying scatters. Jokes were generated and evaluated using the same Llama-3.1-8B-Instruct model.

Given this similarity matrix between pairs of LLM-generated text, we can approximate the proposal distribution of MCMC as $q(y_j|y_i) \approx S_{ij}/S_{i,:}$ where $S_{ii} = 0$. Using this distribution, we can initialize from any $y_i$ and randomly propose a new $y_j$. Because the sample space is constructed from LLMs' outputs with their preferences towards specific semantics, we add a correction term to ensure semantic uniformity of the space when running Markov chain: $\alpha_{bert}(y_j, y_i) \approx S_{ij}/S_{j,:}$.

With the proposal distribution defined, we introduce the critical step of integrating the LLM's choice as the acceptance function in the Metropolis–Hastings algorithm (Zhu et al., 2024a). Specifically, the LLM is prompted to make a binary choice between two candidate passages of text, $y_i$ and $y_j$. This yields the following acceptance function:

$$\alpha_{llm}(y_j, y_i) = \frac{\pi_\theta(y_j|x, \rho_{ij})}{\pi_\theta(y_i|x, \rho_{ij}) + \pi_\theta(y_j|x, \rho_{ij})} \tag{6}$$

Taken together, the detailed balance of the Markov chain can be guaranteed:

$$\pi_\theta(y_i|x, \rho_{ij})q(y_j|y_i)\alpha_{bert}(y_j, y_i)\alpha_{llm}(y_j, y_i) = \pi_\theta(y_j|x, \rho_{ij})q(y_i|y_j)\alpha_{bert}(y_i, y_j)\alpha_{llm}(y_i, y_j) \tag{7}$$

When applied to eliciting the LLM's evaluation of joke funniness, we initialized the Markov chain with a random joke $y_i$. Using the corrected proposal distribution, a candidate joke $y_j$ was then proposed. The LLM was prompted to choose between $y_i$ and $y_j$ based on funniness, with the selected joke retained and the unselected one discarded. A new candidate was subsequently proposed from the corrected distribution, and this process continued until the chain converged. After convergence, the sequence of selected jokes can be interpreted as samples from the LLM's internal representation of a marginalized plan $\pi_\theta(y|x, \rho)$, thereby providing a more reliable ranking of output quality.

**Human evaluations of LLM-generated jokes**. To evaluate whether spMCMC better aligns with people's subjective judgments of joke funniness compared to other self-evaluation methods, we conducted the following experiment. In total, 237 unique jokes were generated by the Llama-3.1-8B-Instruct model during the random generation stage (see examples of jokes in Appendix D), forming the support space for all LLM-based evaluation methods (i.e., $\mathcal{Y} = \{y_1, y_2, \ldots, y_{237}\}$). Running spMCMC with the same model (see prompts in Appendix A.3) for 5000 iterations yielded an updated distribution over $\mathcal{Y}$, with some jokes being selected more frequently than others. For comparison, we also evaluated two additional self-evaluation methods using the same LLM: (i) direct rating and (ii) importance sampling. For direct rating, each joke in $\mathcal{Y}$ was assigned an integer score between 1 and 7 (see Appendix A.3 for detailed prompts). For importance sampling, the LLM was instead prompted to make a binary judgment of whether each joke was funny or not (see Appendix A.3 for detailed prompts).

In summary, the three self-evaluation methods differ in how they score jokes: direct rating assigns an explicit score to each joke, whereas importance sampling and spMCMC rely on the frequency with which a joke is selected as funny. To assess how well different approaches align with human judgments, we conducted an online experiment, recruiting 100 participants from Prolific platform

(50 males and 50 females who are fluent English speakers). Three participants were excluded from the following analysis due to failure in completion or in the attention check. Each participant rated 50 jokes on a 7-point scale of funniness. In total, each joke received 15-25 independent ratings, and the mean score of each joke was taken as its human-assessed funniness. Participants were compensated at a flat rate of £9 per hour.

**Results.** The relationship between spMCMC-derived joke frequency and human-assessed funniness of the jokes is shown in Figure 5. The results show a significant positive correlation (Pearson's $r = 0.23, p < .001$) between the two, indicating that jokes preferred by spMCMC are also more likely to be rated as funnier by humans. Notably, both spMCMC and human evaluation converge on the same joke as the funniest (see the red star in Figure 5), while also agreeing that the joke produced via greedy decoding with maximum token-level likelihood (indicated by the yellow star in Figure 5) is not the funniest at the sequence level. As shown in Table 1, spMCMC, by marginalizing over the latent plans for joke funniness, outperforms importance sampling and direct rating in identifying the jokes rated most highly by humans.

Table 1: Comparison of evaluation methods for rating the funniness of self-generated jokes.

| Methods | Pearson's r with human ratings | Identified funniest joke |
|---|---|---|
| spMCMC | $r = .23$ ($p < .001$) | $1^{st}$ in human ratings |
| Importance Sampling | $r = .07$ ($p = .263$) | $141^{st}$ in human ratings |
| Direct Rating | $r = .13$ ($p = .041$) | $59^{th}$ in human ratings |

## 5 DISCUSSION

In this study, we proposed a Bayesian model of token generation to explain LLMs' capacity for self-improvement through the lens of plan–generation interaction. Our results suggest that while LLMs can form coherent forward-looking plans, their initial token generations are biased. However, as more self-generated tokens are incorporated into the context, contextual uncertainty decreases, leading subsequent generations to better align with the updated latent plan. Over time, the process converges to a token distribution that adheres to a stable plan. Building on these results, we extended the MCMC-with-LLM method to account for the high uncertainty of initial latent plans in open-ended text production through self-play MCMC. Empirical results demonstrate that spMCMC more effectively identifies reward signals in self-evaluation compared to greedy decoding and other self-evaluation approaches.

The proposed Bayesian model provides an explanation for several empirical observations. Updates in latent plans account for the short-horizon planning behavior (Men et al., 2024), where deviations from the plan accumulate and weaken its predictive power over future generations. The prior-driven mismatch between planning and generation at the beginning of a sequence offers an alternative interpretation of Bachmann & Nagarajan (2024), who observed that models often fail to plan effectively at the first step. Furthermore, the planning dynamics observed in the random generation task align with the finding that information about the future is enriched in early layers (Men et al., 2024), and that planning signals are particularly strong at structurally meaningful but semantically minimal tokens such as newline characters before generation (Pochinkov et al., 2024; Lindsey et al., 2025) or commas before numeric samples in the present study. The convergence towards a stable plan during generation also explains why extended reasoning can improve LLM outputs (Wei et al., 2022).

Self-play MCMC offers a promising avenue for further research into LLMs' latent plan and generation. By leveraging the model's own stochasticity and comparative judgments, we can gain insights into how LLMs plan ahead in high-dimensional space. Future research will look further into how to accurately extract the initially planned generation, which is well calibrated to human language, and tap into the potential of using such a Markovian process as an efficient self-improvement paradigm.

## ETHICS STATEMENT

Some of the LLM-generated jokes may contain explicit and/or offensive content, but were not screened out so as to provide a representative comparison. Our human study was approved by

the University of XXX Institutional Review Board (IRB: XXX). All participants provided informed consent prior to participation.

## REPRODUCIBILITY STATEMENT

We provided complete descriptions of our proposed methods in the paper. All experiments were conducted with standard Python packages from Together AI and Hugging Face. The code and human data will be released publicly upon acceptance of the paper.

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

## A PROMPTS

### A.1 PROMPTS USED IN THE HEIGHTS GENERATION TASK OF EXPERIMENT 1

*You are simulating guesses of adult heights (in centimeters) for individuals randomly drawn from the UK population. In each round, output a single integer as your guess for the random person's height. Continue producing guesses one after another, separated by commas, with no explanations or extra text: starting_point,*

### A.2 PROMPTS USED IN THE GAUSSIAN SAMPLING TASK OF EXPERIMENT 2

**System prompt**
*We are sampling integers from a distribution. You will see some samples that has already been drawn from this distribution and your task is to continue sampling integers from this distribution, separated by commas, with no explanations or extra text.*
**User prompt**
*Please continue the sampling: 23, 45, 12, 43, 55, 4, ...,*

### A.3 PROMPTS USED IN JOKE CREATION AND SELF-EVALUATION

Random Joke Creation:

**System prompt**
*You are a master of humor. Your task is to write hilarious, creative and intelligible phrases based on the given prompt. Deliver only the punchline, no setup or explanation.*
**User prompt**
*A complete humorous story within 30 words that will make a lot of people laugh:*

spMCMC:

**System prompt**
*You are a master of humor. Your will see two jokes and your task is to select one which is a better answer to the query 'A complete humorous story within 30 words that will make a lot of people laugh:'. Your response must be a single character: either 'A' or 'B', with no additional text or explanation.*
**User prompt**
*Which is better?*
*A: $\mathcal{J}_i$*
*B: $\mathcal{J}_j$*
*Your selection (A/B):*

Importance Sampling:

**System prompt**
*You are a master of humor. Your will see a joke and your task is to answer whether it is funny or not. Your response must be 'yes' or 'no', with no additional text or explanation.*
**User prompt**
*Given following joke:*
*$\mathcal{J}_i$*
*Your answer (yes/no):*

Direct Rating:

**System prompt**
*You are a master of humor. Your will see a joke and your task is to rate it on a 7-point likert scale, where 1 indicates 'not funny at all' and 7 indicates 'extremely*

*funny'. Your response must be a single number, with no additional text or explanation.*

**User prompt**

*Rate the following joke:*

$\mathcal{J}_i$

*Your rating (1-7):*

## B LLM-GENERATED SAMPLES OF HUMAN HEIGHTS

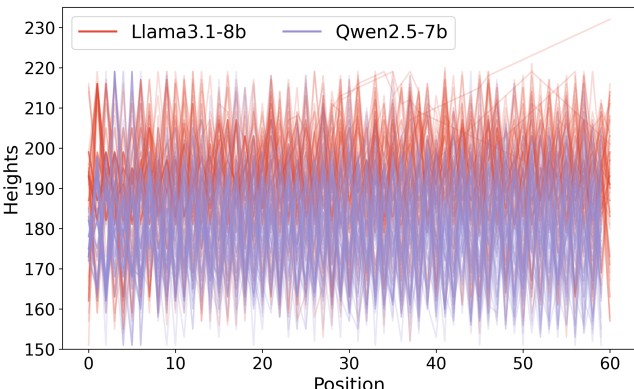

Figure 6: Sequences of height estimations from (red) Llama3.1-8B-Instruct and (purple) Qwen2.5-7B-Instruct.

## C STATISTICAL ANALYSES OF STARTING BIAS IN EXPERIMENT 2

To quantify the starting bias shown up in Experiment 2, we compared samples produced at the first position between the sequences from Gen. I (conditioned on Gaussian) and Gen. II (conditioned on self-generated samples) in each $\mu$ condition. As shown in Table 2 and 3, when conditioned on Gaussian, the LLM presented significant starting bias under more conditions than conditioning on self-generated samples, supporting the prediction of Bayesian token generation model that conditioned on self-generated context is able to reduce the starting bias induced by out-of-distribution context.

Table 2: Comparison of the first generated samples from Llama-3.1-8B-Instruct

| Conditions | Gen. I | Gen. II | Gen. I vs. $\mu$ | Gen. II vs. $\mu$ | Gen. I vs. Gen. II |
|---|---|---|---|---|---|
| $\mu = 50$ | 43.70 (1.115) | 49.24 (.931) | t=-5.65 (p<.001**) | t=-.82 (p=.416) | t=-3.78 (p<.001**) |
| $\mu = 30$ | 29.41 (.955) | 33.68 (1.099) | t=-.62 (p=.540) | t=3.35 (p=.001*) | t=-2.94 (p=.004*) |
| $\mu = 10$ | -1.91 (1.044) | 9.39 (1.159) | t=-11.41 (p<.001**) | t=-.52 (p=.602) | t=-7.26 (p<.001**) |
| $\mu = 0$ | -9.63 (.532) | -2.71 (.641) | t=-18.08 (p<.001**) | t=-4.22 (p<.001**) | t=-8.37 (p<.001**) |
| $\mu = -10$ | -14.53 (.527) | -14.28 (.907) | t=-8.60 (p<.001**) | t=-4.72 (p<.001**) | t=-.25 (p=.802) |
| $\mu = -30$ | -31.84 (.665) | -34.29 (.810) | t=-2.76 (p=.007*) | t=-5.30 (p<.001**) | t=2.35 (p=.020*) |
| $\mu = -50$ | -47.67 (.865) | -49.84 (.924) | t=2.69 (p=.008*) | t=.17 (p=.863) | t=1.72 (p=.088) |

Table 3: Comparison of the first generated samples from Qwen-2.5-7B-Instruct

| Conditions | Gen. I | Gen. II | Gen. I vs. $\mu$ | Gen. II vs. $\mu$ | Gen. I vs. Gen. II |
|---|---|---|---|---|---|
| $\mu = 50$ | 49.23 (.494) | 50.50 (.619) | t=-1.57 (p=.120) | t=.81 (p=.422) | t=-1.63 (p=.105) |
| $\mu = 30$ | 27.10 (.556) | 28.98 (.880) | t=-5.22 (p<.001**) | t=-1.16 (p=.248) | t=-1.87 (p=.063) |
| $\mu = 10$ | 8.51 (.694) | 10.21 (.922) | t=-2.14 (p=.034*) | t=.23 (p=.818) | t=-1.50 (p=.136) |
| $\mu = 0$ | -4.35 (.544) | -3.24 (1.040) | t=-7.99 (p<.001**) | t=-3.12 (p=.002*) | t=-.98 (p=.328) |
| $\mu = -10$ | -14.26 (.529) | -13.04 (.672) | t=-8.06 (p<.001**) | t=-4.53 (p<.001**) | t=-1.45 (p=.150) |
| $\mu = -30$ | -29.52 (.566) | -29.12 (.949) | t=-.84 (p=.402) | t=-.92 (p=.355) | t=-.383 (p=.702) |
| $\mu = -50$ | -50.86 (.522) | -50.37 (.658) | t=-1.64 (p=.104) | t=-.56 (p=.580) | t=-.593 (p=.554) |

## D    EXAMPLES OF JOKES GENERATED BY LLMS

Table 4 shows 20 example jokes created by Llama-3.1-8b-Instruct. Even though these jokes looks diverse, the starting words are highly constrained (8 out of 20 sequences started with the 'After' token), which aligns with our Bayesian model of token generation that predicts a stronger prior bias at the beginning of the sequence.

Table 4: Example jokes randomly generated by Llama-3.1-8B-Instruct

| Index | LLM-generated jokes |
|---|---|
| 1 | One day, a man walked into a library and asked the librarian, "Do you have any books on Pavlov's dogs and Schrödinger's cat?" The librarian replied, "It rings a bell, but I'm not sure if it's here or not." |
| 2 | **After** eating a ghost pepper, John's fart was so hot, it started its own YouTube channel, signed a record deal, and has since won five Grammys. |
| 3 | **After** years of training, a man finally broke the world record for eating a sauerkraut-topped pizza while reciting Shakespeare, only to be dethroned by his hungry cat. |
| 4 | Why did the man bring a ladder to the party? Because he heard the drinks were on the house, literally. |
| 5 | **After** a few drinks, a sarcastic bartender quipped, "That's $12 for beer, $15 for regret, and $20 for your friend saying, 'I told you so'." |
| 6 | A chicken named Chuck walked into a library, saw a turkey juggling books in the non-fiction section and whispered, "Fowl play isn't encouraged here." |
| 7 | Patient brings in a chicken to the doctor and says, 'Doc, I think I'm cracking under the pressure!' The doctor replies, 'Don't worry, it's just a fowl mood.' |
| 8 | **After** realizing he was a chicken, a puzzled man exclaimed, 'Fowl play!' and promptly pecked at his scientist's wallet, shouting, 'Egg-tract me, please!' |
| 9 | **After** ramming into his 'World's Okayest Driver' license plate, he finally got the highway into his garage, and a ticket for absolutely crumpling the whole situation. |
| 10 | A mathematician walked into a bar and ordered a beer, but the bartender asked for his income tax as a tip, because 1 beer + 20% tip = no free drinks. |
| 11 | I told my wife she should face the music, but she said no, she's looking for a 'pain-free divorce', which led to a sonically unsuccessful breakup. |
| 12 | I went to a restaurant and the sign said, 'Breakfast Anytime.' So I ordered French toast during the Renaissance. |
| 13 | **After** trying every diet, he finally lost 20 pounds to a hungry dragon that lived in the fridge, and its fiery breath was his secret sauce. |
| 14 | **After** years of searching, I finally found my long-lost soulmate – unfortunately, it was just me, wearing a backpack till morning alone, at the gas station. |
| 15 | Why did the hipster burn his tongue? He drank his coffee before it was cool. |
| 16 | I once knew a magician named Bob who was so bad at his job, his vanishing acts required police returns and Amber Alerts. |
| 17 | **After** years of research, a scientist finally cracked the code to make chickens binge-watch Netflix, but only if they paid their dues in egg-cellent quality content! |
| 18 | Once, a man walked into a library and asked the librarian, "Do you have 'Chicken Invisible'?" She said, "It's between the Chicken Run books." |
| 19 | A chicken and a penguin walked into a therapist's office, the therapist looked up and said, "Clucking what kind of problems do you guys have?" |
| 20 | At the therapist's office, a man said, 'Doc, I'm addicted to placebos.' The doc smiled and said, 'I think we can make a pill out of that. |