# OpenReview forum: "Deciphering Self-Improvement: Large Language Models Can Take False First Steps"
_ICLR.cc/2026/Conference — Submitted to ICLR 2026_

### Official Review · Reviewer_E2sv · 2025-10-27

**Soundness:** 2
**Presentation:** 3
**Contribution:** 2
**Rating:** 4
**Confidence:** 3

**Summary:**

The paper argues that LLM self-improvement works by exploiting the gap between models’ latent sequence-level plans and the realized outputs. Autoregressive model is reformulated as a Bayesian model to separate model planning and localized inference.

While the OOD user query introduces uncertainty in token-level prediction, the uncertainty decreases as model generation converges to its latent plan, producing reliable reward signal in self-evaluation. A novel method is introduced to marginalize over latent plans for model self-evaluation.

**Strengths:**

- The paper provides a strong motivation for investigating LLM self-improvement.
- The central thesis that self-improvement arises from the gap between a latent plan and a biased initial token realization is novel and intriguing.
- The proposed spMCMC is an intuitive method derived from the theoretical insight supported by experiments covered in 4.1 and 4.2.

**Weaknesses:**

- LLM output is limited to one-dimensional space in the first two experiments. Although they provide some intuitions, these experiments may not reflect the general planning capability of LLMs, e.g. in CoT reasoning.
- Human evaluations are prone to bias and variability. For example, I find the #1 greedy decoding joke more subtle than the funniest joke as ranked by spMCMC.
- Humans, spMCMC, importance sampling, and direct rating have inherently different ranking procedures. For example, spMCMC uses pairwise comparison, importance sampling uses binary evaluation, while humans and direct rating use a 7-point Likert scale. The differences in these procedures may affect the final correlation score.
- spMCMC is validated on only the joke dataset, which does not provide compelling evidence of whether it would work in other settings.

**Questions:**

- In figure 3 (left), why is there a significant drop in $R^2$ at around 40?
- Why use a planning horizon of 8? Is there any specific reason for choosing this number?
- Figure 4 and 6 are too compact and dense, making it difficult to interpret.

---

### Official Review · Reviewer_parA · 2025-11-01

**Soundness:** 3
**Presentation:** 2
**Contribution:** 2
**Rating:** 4
**Confidence:** 3

**Summary:**

This paper aims to provide a mechanistic explanation of LLM self-improvement from a Bayesian perspective.
It proposes a Bayesian framework in which early token generations are biased toward unconditional priors. In contrast, later tokens converge toward stabilized latent plans—offering a principled explanation for why iterative self-evaluation improves output quality.
To operationalize this, the authors introduce spMCMC, a self-play Markov Chain Monte Carlo method designed to elicit reliable reward signals for self-improvement in open-ended text generation.

**Strengths:**

- Presents an elegant and interpretable Bayesian framework for understanding how autoregressive LLMs can self-improve without external supervision.

- Draws insightful parallels between human self-evaluation and LLM self-monitoring, bridging cognitive and computational perspectives.

**Weaknesses:**

- Core findings rely on toy 1D numeric tasks and a small joke-generation dataset, raising concerns about generalization to real-world tasks (e.g., reasoning, summarization).

- Since human ratings may vary, please describe the human evaluation guidelines and annotation procedures in more detail to ensure reproducibility.

- Running 5,000 MCMC iterations for one task is computationally expensive; it remains unclear how the method scales to longer or multi-turn text generation.

**Questions:**

- How well do the findings generalize beyond toy numeric tasks and humor generation to complex reasoning or summarization tasks?

- Can the proposed “latent plan” variables be directly visualized or estimated, for instance via probing or layer-wise attribution?

- How do you determine when the MCMC chain has converged in practice?

- Please discuss the sampling efficiency.

---

### Official Review · Reviewer_Z24N · 2025-11-01

**Soundness:** 2
**Presentation:** 3
**Contribution:** 2
**Rating:** 4
**Confidence:** 5

**Summary:**

The paper proposes a Bayesian account of self-improvement in autoregressive LLMs. The core claim is that models form latent plans in intermediate representations before emitting tokens. Early in decoding, because of context scarcity or prompt–output distribution mismatch, the model relies more on prior token frequencies, creating a transient gap between the latent plans and the actual next-token distribution. As generation unfolds, the emitted token distribution progressively aligns with the latent plans, which constitutes the model’s self-improvement. Building on this view, the paper introduces self-play Markov Chain Monte Carlo (spMCMC) to produce intrinsic rewards for open-ended text generation without external scorers, and reports experiments on three tasks to support the framework.

**Strengths:**

The work offers a conceptually clear and novel explanation for an intrinsic mechanism of self-assessment and self-improvement in LLMs. It grounds the explanation in Bayes’ rule and operationalizes it through the spMCMC method, creating a concrete algorithmic instantiation of the proposed theory.

**Weaknesses:**

There is a mismatch between a stated theoretical expectation and at least one reported empirical pattern. At line 269, the paper asserts that “the strength of planning decreases as the offset increases,” yet in the right panel of the relevant figure, the layer‑25 trace shows R² increasing when the offset exceeds 140. This appears to contradict the hypothesis and is counterintuitive; the manuscript does not discuss this discrepancy.

The evaluation of spMCMC is narrow. First, the benchmark consists of only 237 jokes generated by Llama 3.1 8B Instruct, which is too small to be persuasive. Beyond a “tell a joke” prompt, there are many existing proxies for human evaluation, such as code generation or summarization, that would also be suitable for assessing spMCMC. Second, the baselines are limited to two direct prediction strategies. It remains unclear how spMCMC compares with training a small reward model under similar conditions.

Figures could be clearer. In the main text Figure 2 and Appendix B Figure 6, plotting all raw traces makes patterns hard to read. In Figure 1, the layer‑10 and layer‑15 curves are difficult to discern; applying a logarithmic scale to the vertical axis may improve readability. For Figure 6, consider reporting summary statistics such as means in addition to the full distributions.

**Questions:**

In the first stage of spMCMC, you compute pairwise joke similarities with a Sentence-BERT model (see around line 375). Which specific sentence-transformer variant did you use? Given that sentence transformers can underperform on long-text similarity, did you measure accuracy or reliability for this step, and how sensitive is the second stage of spMCMC to errors here?

---

### Official Review · Reviewer_b16a · 2025-11-03

**Soundness:** 2
**Presentation:** 2
**Contribution:** 2
**Rating:** 2
**Confidence:** 4

**Summary:**

The authors aim to study how LLMs self-improve over the course of autoregressive text generation. They present a Bayesian theory of how LLM text generation proceeds through a combination of locally coherent next-word prediction and longer-horizon planning. Using three experiments, two with generating sequences of random numerical values and one with generating jokes, the authors present evidence supporting their theory where local prediction is traded off against planning. The authors then present a new method for MCMC with LLMs, which is designed to more effectively marginalize over latent plans in an LLM, and show that this significantly boosts performance in the joke generation domain.

**Strengths:**

- The general framing is compelling, self-improvement in text generation, and how LLMs respond differently to user-input text - which may be OOD for the LLM - and self-generated text.
- Formalizing the trade off between local coherence and planning as a Bayesian theory seems like a promising approach, and to my knowledge a novel contribution in this context.
- The authors' proposed method spMCMC seems to produce better results using black box LLMs, which could be generally useful if this is better than alternatives like majority voting.
- Many of the results seem interesting, although I'm not sure if/how they support the authors' main points.

**Weaknesses:**

- Many of the authors' key assumptions around planning are not argued convincingly, which makes it hard for me to get on board with their model and experimental setup.
	- For example, a key argument motivating many of the experiments is "the dynamics can be driven by an asymmetry: the model processes the query, x, differently from its own responses, y<t". The evidence that they provide is that LLMs have a "self-preference bias"
	- The authors claim that an LLM autoregressively generating text becomes "progressively more confident about which tokens are compatible with the current context". What is the evidence for this?
	- The experiments mostly use greedy decoding "to maximize planning behavior", but what evidence is there that greedy decoding maximizes planning?
- The model itself makes big assumptions about the independence of latent planning from the context h_t, and these assumptions are not clearly motivated.
- I'm not entirely sure why sequences of randomly sampled numbers from some distribution is a good domain to study planning.
	- Relatedly, I had trouble making sense of Figure 1, and "177, 174, 187, ... seems a fair plan" seems more like a distribution over tokens rather than a plan.
- The "biasing-then-debiasing" phenomenon was not explained very clearly, which made it difficult to interpret the authors' results, since this is such a central point in results and model predictions.

**Questions:**

- Shouldn't the probability of the plan \rho_t be conditional on the context representation h_t? How are latent plans triggered in the model if not by the context (x, y_{<t})?
- What is the evidence that we should consider the unconditional unigram prior p(y_t) in isolation, rather than a more complicated n-gram prior on tokens?
- In Fig. 1, what's the difference between "188, 194, 187, ... seems a fair plan" and "Not sure what to put here, numbers around 166 feel like good fits, so I say: 180"? Why is the first quote a kind of plan, but the second quote not a plan?
- In Fig. 2 (right) what's going on with qwen having higher correlations longer into the sequence? (left hand side of plot) Does this go against the interpretation that "the strength of planning decreases as the offset increases, indicating that the plan shifts during generation."?
- The new method spMCMC and the results in the third experiment seem promising, but this feels very different from experiments 1 and 2. I had trouble connecting these two sets of domains (random number generation and jokes), and I'm also not sure why generating a series of jokes is a good case study of "planning" - why should one joke's humor depend on a previous joke sample? Can the authors shed some light on the connection between these experimental domains?
- For the claim that LLMs become "progressively more confident about which tokens are compatible with the current context", is this at odds with work such as [1], which seem to show that models do not become progressively more confident as more tokens are generated?
- I have some issues with the second paragraph in the introduction
	- For the claim that human language comprehension is easier than production "because production requires coordination of complex biological systems beyond merely perception". Is there any evidence for this, other than the Broca's area paper from 1861?
	- For the claim "there is no inherent reason to expect LLMs to evaluate better than they generate, since both processes rely on the same neural information processing system", how do the authors square this with the general perspective that discrimination is often a computationally easier problem compared to generation?  E.g. a model that classifies images of bananas vs. oranges could do so with a "mean RGB value" feature, whereas generating these images would require more

[1] Bigelow, E., Holtzman, A., Tanaka, H., & Ullman, T. (2024). Forking paths in neural text generation.

---

### Meta-Review · Area_Chair_Bdni · 2025-12-21

**Summary:**

The paper studies the mechanisms that allows LLMs to self-improve.  The reviewers expressed the following concerns:

1. Key assumptions around planning are not argued convincingly
2. It is not clear that why the random sequence and joke testbeds are suitable to study planning
3. The is a mismatch between the theoretical expectations and the empirical results
4. Core findings rely on toy 1D numeric tasks and a small joke-generation dataset, raising concerns about generalization to real-world tasks (e.g., reasoning, summarization).
5. Human evaluations are prone to bias and variability.

In the absence of any rebuttal or revised paper, the concerns remain.  The paper is not ready for publication.

**Reviewer Concerns:**

In the absence of any rebuttal or revised paper, the concerns remain.

**Reviewer Scores:**

In the absence of any rebuttal or revised paper, I expect the reviewers to maintain their scores and recommend rejection.

---

### Decision · Program_Chairs · 2026-01-26

Reject